# Determination of Permissible Load in Selected Parts of the Human Musculoskeletal System While Feeding Cows with Maize Silage

**Łukasz Kuta [1,\*], Roman Stopa [2], Piotr Komarnicki [2], Monika Słupska [2] and Kamil Górecki [1]**

1    Institute of Environmental Protection and Development, The Faculty of Environmental Engineering and Geodesy, Wrocław University of Environmental and Life Sciences, pl. Grunwaldzki 24, 50-363 Wrocław, Poland; kamilos19944@wp.pl

2    Institute of Agricultural Engineering, The Faculty of Life Sciences and Technology, Wrocław University of Environmental and Life Sciences, 50-375 Wrocław, Poland; roman.stopa@upwr.edu.pl (R.S.); piotr.komarnicki@upwr.edu.pl (P.K.); monika.slupska@upwr.edu.pl (M.S.)

\*    Correspondence: lukasz.kuta@upwr.edu.pl

**Abstract:** Farmers belong to the group of high risk in terms of developing work-related musculoskeletal disorders. A series of tasks, which are often performed in an uncomfortable position, loads exceeding farmers' physical abilities, as well as high repetition of work movements, all contribute significantly to the development of irreversible changes in the musculoskeletal system. Taking into account the above-mentioned circumstances, this study aimed at workload assessment expressed in maximal voluntary contraction (%MVC) while delivering dairy-cattle feed. In the initial phase of this study, a questionnaire was carried out, based on which a load of individual segments of the musculoskeletal system during work was subjectively assessed. On this basis, the areas of the musculoskeletal system were selected in which a risk of the ailments' occurrence was the highest. These studies were carried out directly on farms, where the surface electromyography (sEMG) method was used. On the basis of obtained results, the permissible human load was determined based on mass of the shovel which was used to load and unload maize silage. The obtained results can be used to ensure safe conditions while performing work with high muscle exertion.

**Keywords:** electromyography; hand; muscle; shovel; silage

## 1. Introduction

Disorders in the musculoskeletal system result often from work performed manually. These ailments are particularly manifested by strain and pain in the arms, spine or in the neck and the intensity of such effects may occur over time. The main reasons are manual loading, lifting, pushing or pulling different weighted loads for various distances [1,2]. Current reports show that musculoskeletal disorders are suffered by nearly 25% of European Union employees [3]. The cause of these symptoms is long-term loads of the musculoskeletal parts of workers due to work in inappropriate conditions.

In agriculture, due to a wide scope of performed activities, this problem requires a deeper recognition in terms of determining the causes and possibilities to minimize the effects of physical overloading of worker's body. A wide range of occupational risk occurs while performing activities on farms [4,5].

Despite the mechanization of agricultural production, the problem of ailments in the musculoskeletal system is still noticeable [6], especially while obtaining the milk. Year by year,

there are more systems in use, which ensure more precise feeding as well as livestock raising method management [7,8].

Based on conducted studies, it was found that a milking process by means of milking pipeline generated a lower load for milkers than corresponding milking into bubbles [9]. The influence on development of disorders in the musculoskeletal system may cause muscle static load, which particularly concerns truck drivers [10,11]. In order to measure with high precision a generated load of musculoskeletal system, Hansson used surface electromyography [12]. The correct measurement was used to design proper workstations [13,14]. An example of this is the studies conducted by [10,15,16] who, based on the analysis of a driver's body pressure on the seat surface, designed a proper seat profile which reduced physical fatigue and discomfort while driving. So far, numerous ergonomic factors have been discovered that influenced the level of occupational safety [17,18].

Precise estimation of the listed factors contributes to accurate determination of the permissible mass of an object which can be transferred in working conditions at a measured repetition frequency [19]. Observation of manual workers behavior allows for constant increase in the level of safety at work by equipping workstations with the necessary elements [20,21].

Jaworski, Lach, Fabunmi and Henk emphasized that a worker's body during lifting accompanied with high repetitiveness of tasks is the most common reason for the appearance of ailments in the human musculoskeletal system, e.g., during manual loading and unloading [22–25].

The lumbar spine, according to Solecki is a part of the musculoskeletal system most exposed to pain [26]. It results from force generated between the vertebrae while loading, e.g., bags with fertilizer or feed and then moving them for a long time. Barrero, Xiang and Hodges confirmed positive correlation between physical load and appearance of ailments in specific segments of the musculoskeletal system [27–29]. The authors stated that there are many professions that require in-depth analysis of the causes of musculoskeletal disorders.

In-depth assessment of both static and dynamic loads on human muscles can be performed using the sEMG. In order to obtain accurate results, some of technical criteria should be met, e.g., precise electrode placement on skin, proper preparation of the worker's skin or accurate operation of the preamplifiers. Both the depolarization of muscle fibers and the neuromuscular junction result in the EMG signal. On basis of this signal a force generated by the muscles can be assessed.

Based on this assessment of the developed tone and strength of muscle, it is possible to determine the chances of muscle injury or pain, including in the area of the lumbar spine [29]. Electromyography is a non-invasive method which enables the assessment of the electrical potential of a working muscle with a view to preventing it from being overloaded during work [30]. The EMG system is frequently used in the rehabilitation of patients to determine the optimal positions of the postures of injured people [31–33].

The use of electromyography in ergonomic studies by Swedish scientists has clearly highlighted the need to change the approach to the way devices, such as milking machines, are designed [12]. In their work, they confirmed that the method of surface electromyography (sEMG) is easy to apply, safe for the person being examined and is a precise method of testing loads experienced by farmers.

On the basis of the measured strength-forming potential of farmers' muscles, it was possible to determine the maximum mass of milking machine, which can be used when milking cows in a "herring bone" milking shed [34,35].

This study aimed at determining the load of the selected musculoskeletal system of farmers while loading, transport and unloading maize silage. Firstly, a survey was conducted among farmers and, based on this, the most common loading tasks according to farmers' opinions were selected. Another purpose was to determine the maximum mass of the shovel so that the level of forearm load involved does not exceed the permissible value (30% MVC) [36].

## 2. Materials and Methods

### 2.1. Participants

The research sample consisted of 50 farmers (18 women and 32 men) who, while performing daily farm work, were exposed to significant loads on the musculoskeletal system. During the research, the farmers did not perform any other physical work which was unrelated to agricultural activity. All farmers had been performed agricultural works at least 3 years. Each of the farmers was informed about how the study was to be prepared, how the measurements should be carried out and what a purpose of the study was, and those who agreed to complete the survey and the EMG measurements signed a statement. The research-group characteristics are presented in Table 1.

**Table 1.** Anthropometric characteristic of the surveyed farmers (N = 50).

| Anthropometric Features | Men (n = 32) | | Women (n = 18) | |
|---|---|---|---|---|
| | Average | Standard Deviation | Average | Standard Deviation |
| Age [years] | 45 | 3.6 | 37 | 4.3 |
| Height [cm] | 177 | 6.1 | 168 | 10.0 |
| Body mass [kg] | 82 | 2.9 | 80 | 4.8 |

### 2.2. Survey

When choosing farmers for the surveys, the basic criterion was that they ran agricultural production. In order to identify the problem of muscle load during manual work performed on a farm, a survey consisting of 5 open questions was conducted. All the questions, addressed to farmers, were the following:

- Which of the farm activities places the greatest load on your musculoskeletal system?
- Which parts of the body are most notably stressed when you perform the activity in question 1?

Please assess on a scale of 1–10 the level of physical load on the parts of the body you gave in question 2.

- Do you feel any pain or discomfort when you perform the activities you gave in question 1?
- How many times a day do you perform the activity that you gave in question 1?

In Figure 1, the places where the farmers' muscle-load measurements are to be made, determined on the basis of the survey, are shown.

Muscle groups selected to these studies were the following; Interosseus, Brachioradialis, Latissimus dorsi, Biceps branchii, and Neck extensors. These parts of worker's body are the most common used during such works.

### 2.3. Task Performance

According to farmers the most common loading task within a farmer's daily work was feeding the maize silage, which included manual loading, manual transport and manual unloading of feed. The subjects of measurements constituted loads in the arms, in the forearms, in the neck, in the spine and in the wrists. The first phase of work was manual loading of maize silage on a single-wheeled wheelbarrow (total capacity 60 kg). The average distance between the silage pile and the cowshed was 20 m. The mean outdoor temperature ranged from 20 °C to 25 °C, humidity 60% (±5%). A hand-held shovel (mass 2 kg without load) was used for this purpose. A wheelbarrow loaded with silage was transported to the feed corridor in the cowshed, where the feed was unloaded manually (Figure 2). Approximately 25 kg of silage per day was transported for each cow (mean 15 wheelbarrows/day

per farm). As farmers noted, this task followed involved extra loads in the musculoskeletal system, especially in the hands and in the spine.

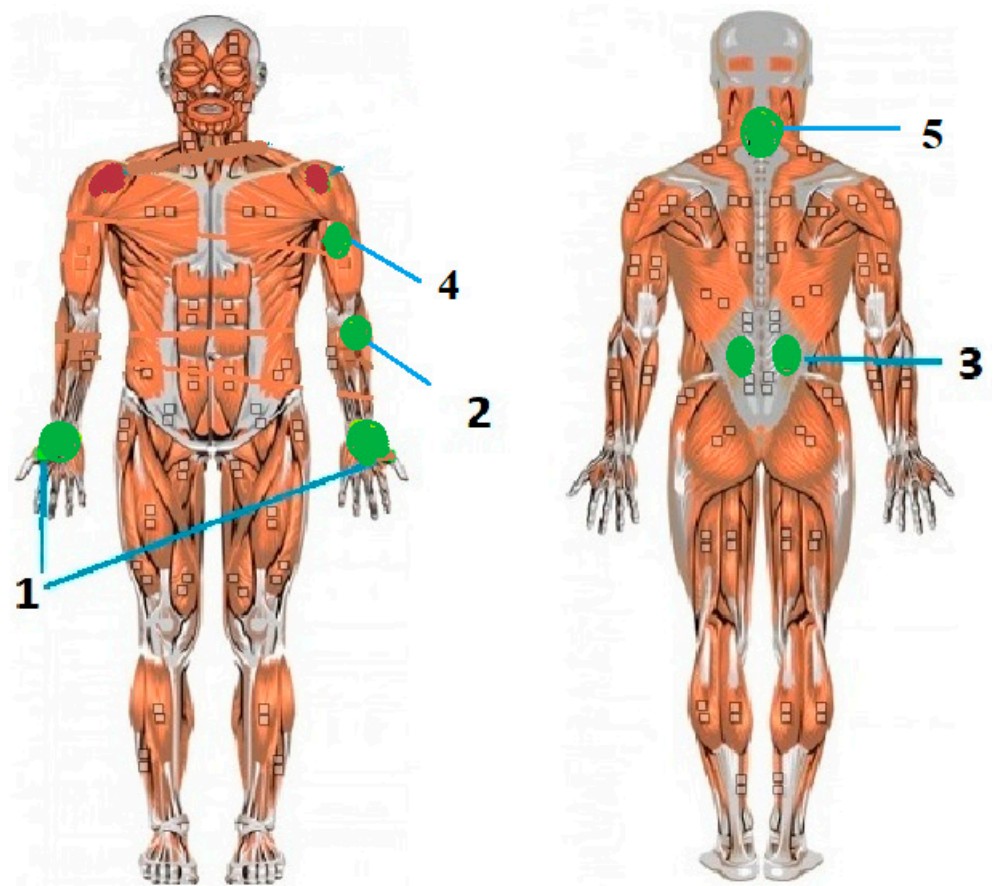

**Figure 1.** Muscle groups of examined farmers; 1—Interosseus, 2—Brachioradialis, 3—Latissimus dorsi, 4—Biceps brachii, 5—Neck extensors.

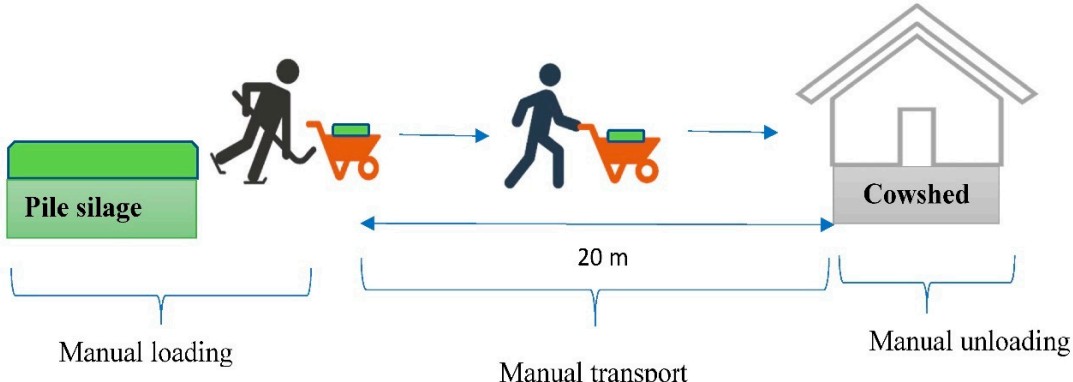

**Figure 2.** Diagram of activities for which measurements were performed.

### 2.4. Muscle Load Measurement

Non-invasive, 4-channel surface electromyography (sEMG) was applied to these tests. On the basis of numerous samples, this system has been awarded with two international certificates, SENIAM and ISEK, for the high quality of given results. The sEMG set consists of electrodes, a preamplifier, Wi-Fi adapter and computer system (display option) (Figure 3). For these studies, a 30 mm × 24 mm dimensioned hydrogel AgCl electrode with a sampling frequency of 1600 Hz was used. The electrodes

were placed directly on the muscle and the distance between them was 2 cm. The preamplifier provides the highest quality of electric signal and it reduces the disturbances. Due to low-impedance output, cable-motion distortion was eliminated. This element was made of medical stainless steel of 10 g in mass. Before the measurement the farmer's skin was washed and cleaned. The mean impedance of the skin was 2.5 kΩ. Fixed frequency range for the EMG preamplifier was 10 Hz for the higher range and 500 Hz for the lower range. The sampling rate was 1000 Hz. Additionally, the Fast Fourier Transformation was used. The smoothing of recorded signal with RMS (Root Mean Square) was defined in the time frame with 50 ms.

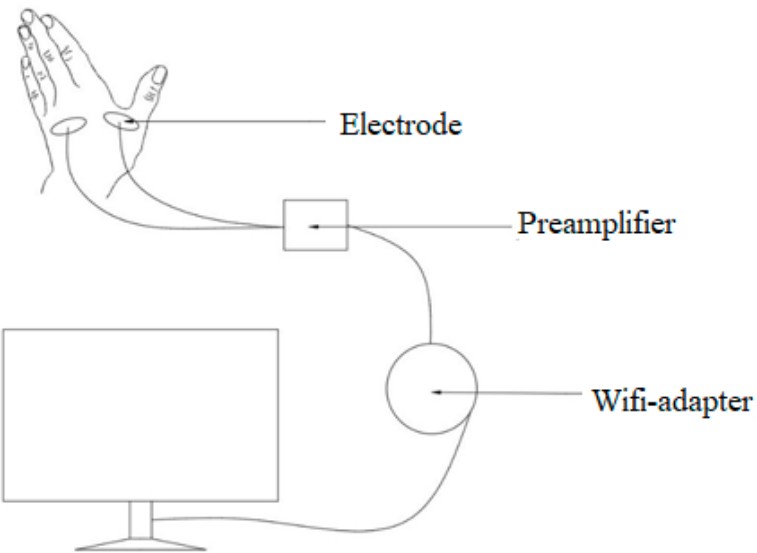

**Figure 3.** The model of muscle tension recording with terms of the surface electromyography (sEMG) system.

Before taking the measurements, an exact location of the given muscle and connection of the cables were determined. Besides, quality of the EMG baseline was checked. The level of EMG electrode impedance was measured too. The electrode pair was placed centrally above the muscle, taking into account the possibility of muscle relocation under the electrode, especially in the biceps brachii.

The results were expressed in milivolts (mV), as well as percentage (%) of Maximal Voluntary Contraction (%MVC) generated by examined muscle. The measurement's error was ±2 mV and the median ranged from 40 mV to +40 mV. During the studies, all the data were stored in an extra disk [36–38].

### 2.5. Statistics

The ANOVA test for independent groups was carried out to analyze the differences between a worker's workload by means of the STATISTICA 12 program.

The ANOVA test was conducted for inspections where there was normality in the distribution of the feature being examined, it was an independent model and where in all populations there was equal variance in the examined variable. If any of these conditions were not met, then the Student's test was conducted. The significance level for this test was 0.05. Moreover, a Levene's test was conducted. From the one hand when the significance of Levene's test is less than 0.05 such difference is significant and the variances are not homogeneous (not similar). From the other hand, while significance of Levene's test is higher than 0.05 the differences are insignificant. The data were presented as mean ± SD.

## 3. Results

In the first step of this study the survey was conducted. In Section 3.1 subchapter the results of this survey were presented.

### 3.1. Survey

On the basis of this survey, it was found that feeding cows with maize silage were performed nearly by 40% farmers. For silage feeding task, farmers individually determined the degree of perceived load in selected parts of their body. When farmers evaluated their body part load in the range 1–4 points it was acceptable, 5–7 points medium loaded and 8–10 points it was overloaded. In order to study the actual EMG load level, segments of the musculoskeletal system were determined for which the farmers' subjective level of was at least 5 points.

From the conducted survey it follows, that the highest loads, according to farmers, were felt in the lumbar spine, forearms, arms, neck and wrists (Table 2). All the mentioned segments in the musculoskeletal system will be analyzed in the next part of this study by means of the sEMG method.

**Table 2.** Subjective assessment of the average load level expressed in range of 1–10 resulting from silage feeding (manual loading, transport and unloading).

| Body Part | Level of Physical Load | Load Level |
|---|---|---|
| Neck | 5.1 | Medium |
| Arm | 7.6 | Medium |
| Forearm | 8.2 | High |
| Lumbar spine | 8.4 | High |
| Wrist | 5.0 | Medium |

### 3.2. Manual Silage Loading

While loading the maize silage on wheelbarrow the highest external load was focused on the forearms, which were exposed to performing numerous repetitions. The spine, inclined forward, was exposed to higher load, especially in the lumbar part. The neck makes repetitive movements to the left or right sides, hence increased discomfort and pain. For instance, in the forearms, where the mean load for women was 29% MVC, among men, it was 27% MVC. The peak value for women was 36% MVC, whereas 34% MVC reached for men.

On the basis of the statistical analysis, the distribution of dynamic load in the forearm was presented in Figure 4. From the student's test it follows that the load = among studied group was not statistically significant; the probability value ($p$) obtained in the student's t-test was $p = 0.36$. This means that average dynamic load in the studied group was very similar.

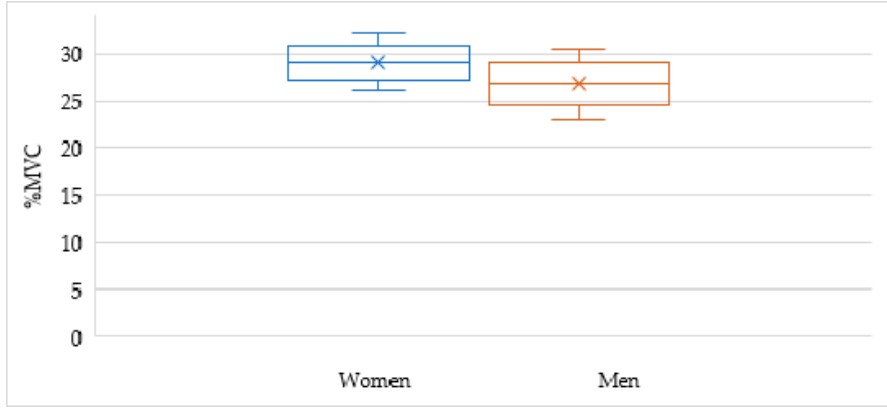

**Figure 4.** A frame-whisker chart of load distribution in the forearms generated by the examined farmers.

The highest acceptable dynamic load in the forearms should not exceed 30% MVC. Therefore, the maximum total mass of the shovel (2 kg) plus the mass of silage was determined on the basis of the forearm muscle load (Figure 5).

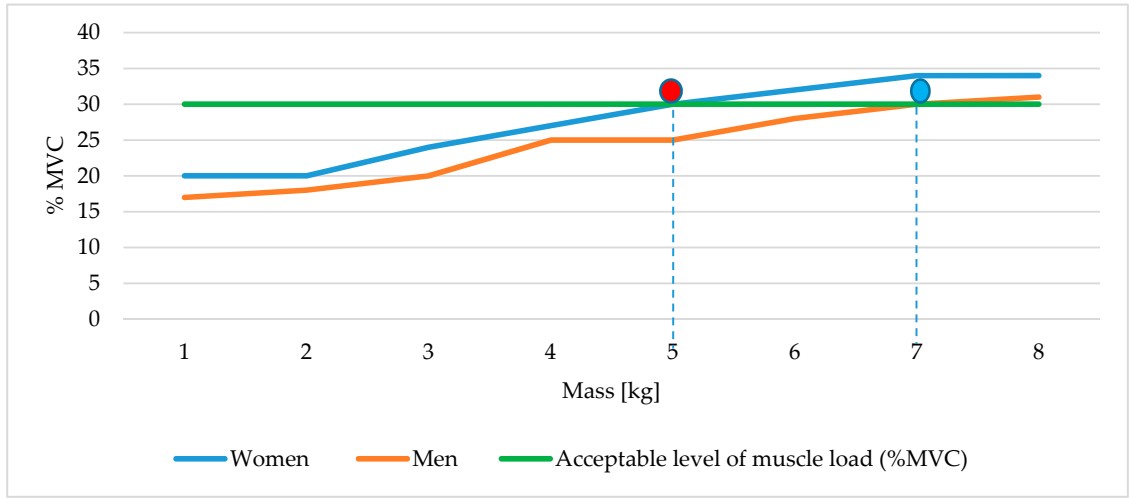

**Figure 5.** Determination of the maximum mass of the handled shovel used while loading (for women—red point and for men—blue point).

The maximum mass of the unit muscle load (points where the green line contacts the red and bluelines) for women was 5.2 kg, while for men it was 7.0 kg, including similar working conditions. The lowest load was in the neck. The peak was recorded in the spine and ranged from 19% to 25% MVC, respectively. The lowest figure was in women's necks, 11% MVC, and for men's wrists, 14% MVC. Figure 6 shows the standard deviation for the given value.

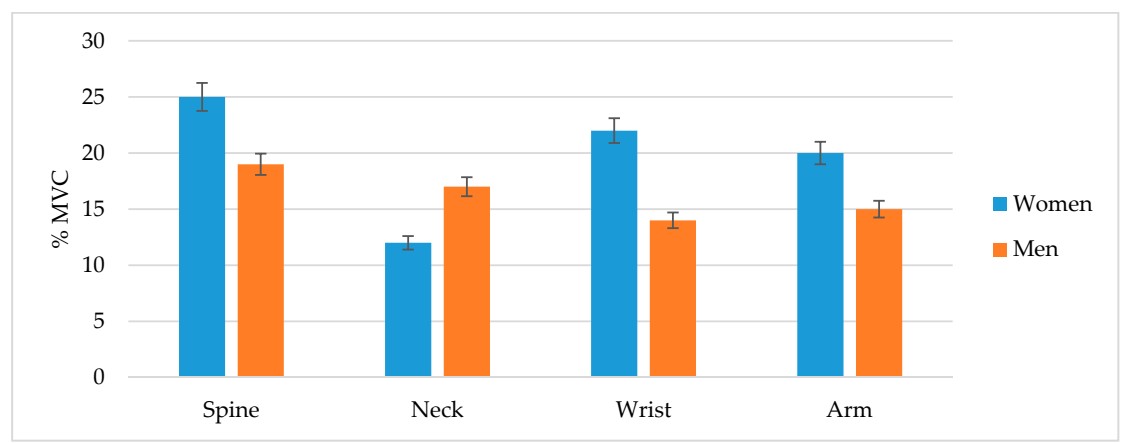

**Figure 6.** Mean muscle load in the in the examined body parts.

### 3.3. Manual Transport and Unloading

During feed transport, the farmer was exposed to static load, which resulted from body posture. For women, mean load values were lower than corresponding values for men and ranged from 32.5% MVC to 39.5% MVC. For men, muscle load reached 32–44% MVC.

Mean load as well as standard deviation were shown in Figure 7. On the basis of statistical analysis, it was found that results obtained in the group of examined farmers did not indicate a significant statistical difference. The probability in the student's t-test was $p = 0.55$, ($p > 0.05$), therefore there were not significant statistical differences between the loads between men and women. Additionally,

in the Levene's test *p*-value was 0.45, hence the differences in the results between women and men were insignificant.

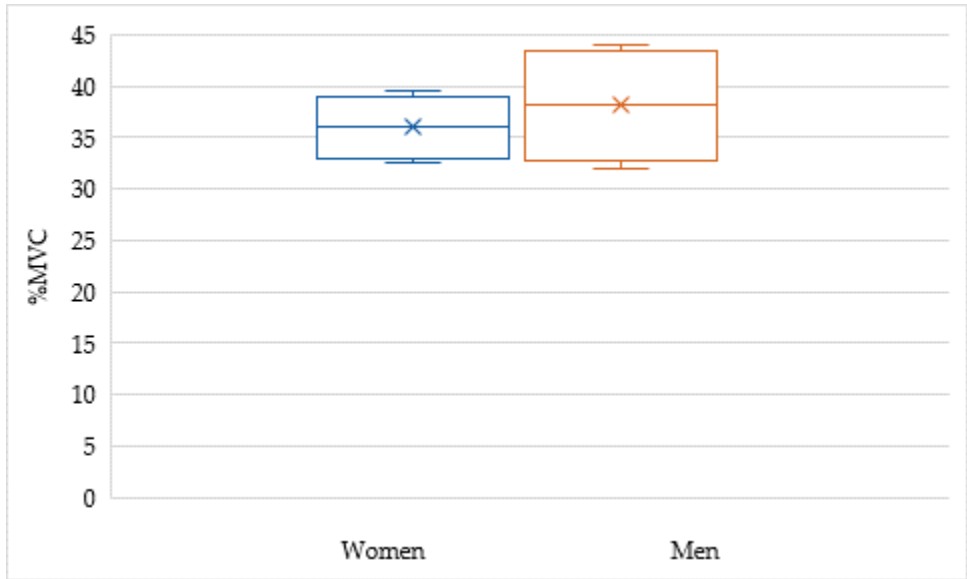

**Figure 7.** Load distribution of the examined farmers groups.

Differences in the obtained findings resulted mainly from the mass of the transported load. Usually women transported smaller unit loads of silage, and as consequence they made more work repetitions. The highest load among men during feed transport occurred in the lumbar spine, approximately reaching 41% MVC, whereas the lowest was in the muscle spine: 17% MVC. Among women the highest farmer's muscle load in the lumbar spine was 44% MVC (Table 3).

**Table 3.** Mean %MVC values of selected farmer's muscle groups during silage transport to the cowshed.

| Body Part | Men | Female |
|---|---|---|
| | MVC [%] | |
| Forearm | 28 | 30 |
| Wrist | 22 | 24 |
| Neck | 17 | 21 |
| Spine | 41 | 44 |
| Arm | 31 | 34 |

As results from the ergonomic norm of static loads, in all cases presented in Table 3 the acceptable values of loads resulting from the maintained farmer's body position were exceeded. Therefore, overloaded muscles could cause disorders. The statistical analysis showed that differences in the generated load were not statistically significant in each group. The probability in the student's t-test was 0.71. Maintaining the body posture during feed unloading was not comfortable because it required the generation of a lot of physical effort. The measurements were made also for other parts of a farmer's musculoskeletal system. The highest recorded values were recorded in the lumbar spine for men at 41% MVC and women at 35% MVC, respectively, (Figure 8). The Levene's test showed insignificant difference between examined groups.

Unloading of silage caused the highest load in the spine. In order to keep the right posture, the external load and gripping the handle of the wheelbarrow both affect the high load in the wrists.

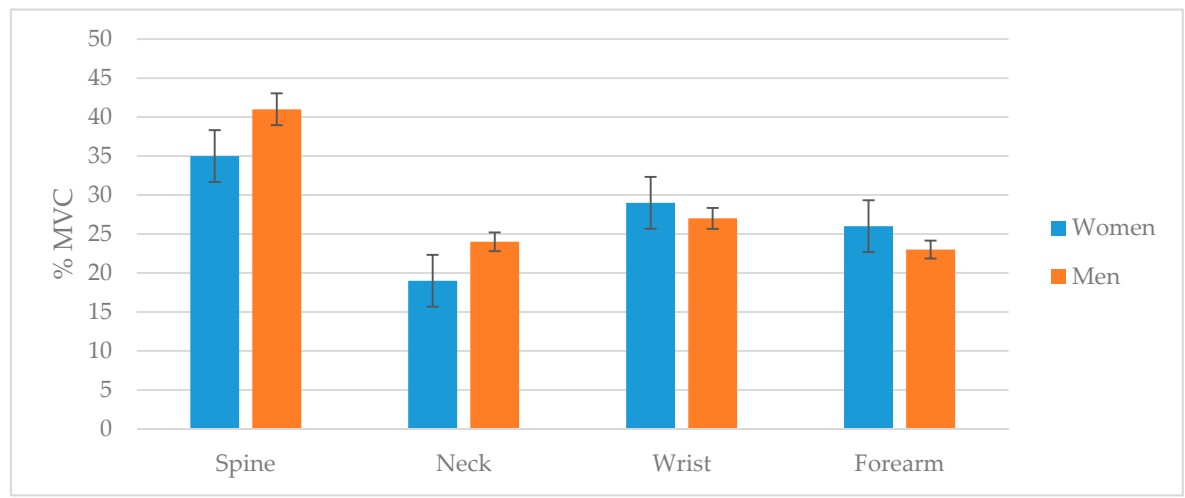

**Figure 8.** Mean muscle load in the during manual unloading of the maize silage.

## 4. Discussion

The use of surface electromyography in order to determine the dynamic load for farmer workers is an innovative approach in the field of ergonomic research, as well as workplace safety among Polish farmers. The indisputable advantage of the method is its high accuracy of measurement in comparison to the tabular or quantitative-qualitative methods that have so far been used. So far, Komarnicki analyzed a load influenced biological material, but in this study, they described a load generated by farmer's muscles during silage feeding [39]. This was the case with milking activities in a herring bone milking shed and when milking with a bubble milking machine. One of the research assumptions in this work was the determination of the maximum unit mass of the load that was being transferred in the process of silage loading. This goal guided the research of scientists in the past, where, for example, a maximum mass of 3 kg was determined for the milking machine in the milking shed [34]. The forearm load values that were obtained, expressed in % MVC, are similar to the results of the loads for a milker while washing, massaging and attaching the milking apparatus, being in the range of 15–25% MVC [34]. Stal emphasized in their scientific papers that a %MVC score above 15% may be a contributory factor in the development of musculoskeletal complaints among employees [14,40,41]. The physical-load results obtained by milkers, in most cases, was in excess of the acceptable level, reaching values above 30% MVC. The authors also emphasized that roughly 50% of all farm workers have complained of pain and discomfort when performing physical activities on a farm.

In the case of manually transporting silage, the forearm load exceeded 30% MVC, hence it can lead to discomfort while working and, consequently, could lead to injury or other complaints. Similar studies were carried out among employees who were handling manual loads of 10–30 kg in production halls. The EMG results showed the load to be in the range of 30%–40% MVC, i.e., similar to the result obtained in the work of farmers who were transporting silage.

The results that were obtained expand the scope of the current state of knowledge in regarding women's physical loads, determining the level of the maximum unit load for repetitive tasks. In addition, in the studies conducted by Zaniewska, 5% of surveyed farmers noted that negative symptoms in the musculoskeletal occurred regularly after physical load [42]. For instance, a work in the construction industry can be classified as one of the most dangerous professions, because of many disorders occurring in the worker's musculoskeletal [43]. Ensuring safety on construction site is one of the most important elements of the entire safety management [44]. Musculoskeletal disorders are one of the most common health problems at work and affect millions of employees every year [45]. So far, the OCRA (Occupational Repetitive Actions) or Strain Index (SI) methods have been used to assess the load associated with the physical work of upper limbs. These methods are based on specific patterns. They allow one to evaluate an occupational risk [46,47]. From the point of view of muscle load, a time

of load, body position, but also work breaks are important. Miedema developed the concept of the optimal position during work in order to reduce muscle fatigue [48]. Yates evaluated some of postures among workers. He defined the most acceptable position at work. From his studies it follows that, right work condition without overloading body parts result in lower risk of musculoskeletal disorders [49]. Generally, they observed workers at work, in comparison to these studies, when the EMG system allows to record real biological signal adequate to examined farmer. Nowadays, the worker's muscles are exposed to different disorders. It depends mainly on the structure of the muscle, load of the muscle or hormonal balance. Therefore, very important issue is to reduce muscle overload especially at work [50].

## 5. Conclusions

This assessment of loads in the human musculoskeletal system is particularly important from the point of view of the possibility of introducing preventive actions in similar positions. With reference to the obtained results, it needs to be stated that, for manual loading of maize silage at a specified frequency of repetition, the maximum mass of the load for women to carry, determined on the basis of the measurements, should be 5.2 kg, and for men, 7.0 kg. The highest load values while loading silage were recorded in the spine area both of women and men: 19% MVC in men and 25% MVC in women. During the process of unloading, the highest load occurred in the farmers' backbone: 35% MVC for women and 40% MVC for men. Subjective assessments of the level of physical load on employees do not differ significantly from the results that were obtained on the basis of load measurement. The risk of ailments in the musculoskeletal system is higher among women than among men for the jobs that were compared.

**Author Contributions:** Conceptualization, Ł.K. and P.K.; methodology, R.S.; software, M.S.; validation, Ł.K., K.G. and R.S.; formal analysis, Ł.K.; investigation, Ł.K.; resources, P.K.; data curation, Ł.K.; writing—original draft preparation, Ł.K.; writing—review and editing, Ł.K.; visualization, K.G.; supervision, Ł.K.; project administration, M.S.; funding acquisition, K.G. All authors have read and agreed to the published version of the manuscript.

**Funding:** This research received no external funding.

**Acknowledgments:** Special thanks to the farmers who supported these studies in their own farms.

**Conflicts of Interest:** The authors declare no conflict of interest.

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
