# Peer review of "Determination of Permissible Load in Selected Parts of the Human Musculoskeletal System While Feeding Cows with Maize Silage"

_applsci, doi:10.3390/app10207125_

Round 1
Reviewer 1 Report
Determination of Permissible Load In Selected Parts of the Human Musculoskeletal System While Feeding Cows with Maize Silage
– Kuta et. al., Applied Science
The authors present an interesting method of determining the maximum load that male and female farmers should bear in efforts to avoid excessive muscle pain and strain during bovine feeding. The authors use survey assessment to determine which farm activities farmers think require the most muscular effort, and which muscles they think are most stressed during these activities. The authors then follow this survey with EMG testing to determine %MVC of specific muscle groups during the activities selected based on the above mentioned survey. This work presents a well stated and succinct report of the findings, with appropriate conclusions drawn concerning maximum recommended load. Several specific points for corrections are listed below. Most notably, expansion of statistical methods to include power analysis is highly recommended. Careful review of the text for correct grammar is essential.
Abstract
Does MVC stand for maximum voluntary contraction? This acronym should be defined before its first usage in the abstract.
Introduction
Line 32 – Should this be “comparison shows…”?
Line 46 – Should this be “with high precision…”?
Line 70 – sEMG was already defined in the abstract, the acronym should be used from this point forward.
Materials and Methods
As table 2 and section 2.2 convey the results of the survey, it is recommended that this section be moved to the results section of the manuscript.
Information should be provided about initial or post hoc power analysis in the statistics section of the materials and methods.
Results
Line 198 – Why should the highest dynamic load in the forearms not exceed 30% MVC? Is this an accepted standard? If so, please cite.
Line 228 – should this be “the lowest was in the neck:”?
Discussion
Line 253 – should this be “farmers’ muscles…”?
Line 267 – “which definitely affected” is very strong language. Please consider revising.

Author Response
Dear Reviewer,
Please see the attachment. I have added the responses to Your comments.
With best wishes
Łukasz Kuta

Reviewer 2 Report
Reviewer Comments:
ApplSci – 937119:
Determination of permissible load in selected parts of the human musculoskeletal system while feeding cows with maize silage.
Kuta et al.
Summary:
This is an interesting and relevant topic, however, the precise sites of measurement and the exact testing and use of sEMG are not documented precisely enough to enable a repeat study. Whilst the findings of such a trial are of potential use in the farming industry, I am not convinced by this particular study, as no details are given about previous injuries in the study group, their level of physical fitness, the use of the sEMG technique or precisely how %MVC was arrived at.
General Comments:
- The language throughout this manuscript would benefit from a read through by a native English speaker.
- Line 47: The use of sEMG is neither precise (it measures both neuromuscular junction and muscle fibre depolarization), and has a number of technical as well as interpretative issues – these should be mentioned – see SENIAM. Besides which sEMG does not measure muscle load, it measures nerve and muscle activity .. at best the sEMG parameters can be plotted against muscle load to give a correlation.
- This study appears to have no testable hypothesis/hypotheses, this is rather worrying.
- What level of physical fitness did the participants have? What was their previous history of injury, and how recently were they injured, out of work? What degree of physical training did they undertake in their free time?
- Figure 1: FCR is a forearm muscle and not at the wrist as shown. Brachioradialis is positioned correctly as is biceps brachii – but was it the long or short head that was measured? What is meant by lumbar spine muscles? The marker indicates latissimus dorsi. Likewise which muscles are identified under the term cervical .. it looks like trapezius has been measured? The authors need to be much more precise in the muscles they have selected for measurement. It would also be useful to have details for the muscles that were measured, origin, insertion, innervation and function.
- Line 160-161: The authors need to describe where on each muscle they placed the electrodes, with sEMG electrode placement is essential as you can easily place electrodes on the neuromuscular junction and record almost no signal .. how did the authors ensure this did not happen?
- Line 170-171: How was %MVC calculated? Did you measure load using a grip test device? What device was used to measure this parameter?
- 5 Statistics: I could use a more detailed explanation of the statistical approach adopted. Were the data normally distributed? If not, what non-parametric test was performed on the data? What level of significance was adopted and how are your data presented (mean +/- SD or SEM?).
- What consideration has been given to the use of the muscles measured. To what extent was muscle contraction performed in a concentric and eccentric manner? How does this relate to the proposed acceptable level of muscle load shown in the figures? The authors need to explain the underlying physiology.
- In general the discussion section is weak and more could be made of the results, the underlying physiology and specific cases of muscle use/over-use.
Specific Comments:
- Lines 63-64: “Many authors” surely this sentence should be backed up with references to these many studies?
- Lines 81-82: I suspect the authors mean a “herring bone” rather than “fish bone” milking shed.
- Line 170: Which results were measured in mV? I assume the sEMG signal, but what about mean signal frequency, RMS, mean signal amplitude, maximum signal amplitude etc?
- Figure 4: This graph should ideally have an axis that runs from zero as there is no significant difference between the %MVC values, despite the graph trying to make it look as though there might be.
- Line 198: 30% MVC as max acceptable dynamic load .. where does this value come from? Reference?
- Figure 6: Were any of these values significantly different from one another .. muscle versus muscle and men versus women? If not then mention this in the figure legend.
- Figure 7: This has the same axis issue as previously mentioned (see Specific Comments 4).
Author Response

(The authors gave the same response as above.)

Reviewer 3 Report
The article has a correct IMRaD structure. The article is interesting and valuable but there isn't a new method in the article so the article isn't intelectuall stimulating. The analyzed problem is not artificial.The problem is clearly defined. The work is not very complicated. The article is important to colleagues working in the field. The number of references is proper and references are modern. The introduction state the purpose of the paper. The aim of work is rationally selected. The survey questions are designed correctly. The experiment is done correctly and the research group is relatively large.The indicated relationships are correct. The scientific value of the work is not very high. Statistical calculus is elementary. Correct conclusions were made. I believe that the raised problem raised has been resolved. The strong point of this research work is its usefulness 1. How do you validate your myography results ? 2. How do you determine 100% MCV ? What motor activity was taken into account? 1. ,,Subjective assessments of the level of physical load on employees do not differ significantly from the results that were obtained on the basis of load measurement." - you should write it in a strict sense. Mandatory change: You should add a number to the end of your affiliation. You sholud improve the quality of figures 4,5,6,7,8. A significant increase in the aesthetics of the figures will make the article attractive to the reader. Optional changes: You should consider removing the Noraxon name from the text.Author Response
Dear Reviewer,
Please see the attachment. I have added the responses to Your comments.
With best wishes
Łukasz Kuta
